# Genome-Wide Association Analysis of Salt-Tolerant Traits in Terrestrial Cotton at Seedling Stage

**DOI:** 10.3390/plants11010097

**Published:** 2021-12-29

**Authors:** Juyun Zheng, Zeliang Zhang, Zhaolong Gong, Yajun Liang, Zhiwei Sang, Yanchao Xu, Xueyuan Li, Junduo Wang

**Affiliations:** 1Economic Crops Research Institute, Xinjiang Academy of Agricultural Science (XAAS), Urumqi 830001, China; zjypp8866@126.com (J.Z.); zzldeyouxiang@126.com (Z.Z.); zhangzeliang8866@163.com (Z.G.); 13999966149@163.com (Y.L.); s2564976727@163.com (Z.S.); 2Engineering Research Centre of Cotton, Ministry of Education, College of Agriculture, Xinjiang Agricultural University, 311 Nongda East Road, Urumqi 830052, China; 3State Key Laboratory of Cotton Biology (China), Institute of Cotton Research, Chinese Academy of Agricultural Science (ICR-CAAS), Anyang 455000, China; b27274092@163.com

**Keywords:** upland cotton, salt stress, SNP, GWAS, salt tolerance gene

## Abstract

Soil salinization is the main abiotic stress factor affecting agricultural production worldwide, and salt stress has a significant impact on plant growth and development. Cotton is one of the most salt-tolerant crops. Therefore, the selection and utilization of salt-tolerant germplasm resources and the excavation of salt resistance genes play important roles in improving cotton production in saline–alkali soils. In this study, we analysed the population structure and genetic diversity of a total 149 cotton plant materials including 137 elite *Gossypium hirsutum* cultivar accessions collected from China and 12 elite *Gossypium hirsutum* cultivar accessions collected from around the world. Illumina Cotton SNP 70 K was used to obtain genome-wide single-nucleotide polymorphism (SNP) data for 149 elite *Gossypium hirsutum* cultivar accessions, and 18,430 highly consistent SNP loci were obtained by filtering. It was assessed by using PCA principal component analysis so that the 149 elite *Gossypium hirsutum* cultivar accessions could be divided into two subgroups, including subgroup 1 with 78 materials and subgroup 2 with 71 materials. Using the obtained SNP and other marker genotype test results, under salt stress, the salt tolerance traits 3d Germination potential, 3d Radicle length drop rate, 7d Germination rate, 7d Radicle length drop rate, 7d Germination weight, 3d Radicle length, 7d Radicle length, Relative Germination potential, Relative Germination rate, 7d Radicle weight drop rate, Salt tolerance index 3d Germination potential index, 3d Radicle length index, 7d Radicle length index, 7d Radicle weight index and 7d Germination rate index were evaluated by GWAS (genome-wide association analysis). A total of 27 SNP markers closely related to the salt tolerance traits and 15 SNP markers closely related to the salt tolerance index were detected. At the SNP locus associated with phenotyping, *Gh_D01G0943*, *Gh_D01G0945*, *Gh_A01G0906*, *Gh_A01G0908*, *Gh_D08G1308* and *Gh_D08G1309* related to plant salt tolerance were detected, and they were found to be involved in intracellular transport, sucrose synthesis, osmotic pressure balance, transmembrane transport, N-glycosylation, auxin response and cell amplification. This study provides a theoretical basis for the selection and breeding of salt-tolerant upland cotton varieties.

## 1. Introduction

Soil salinization is one of the main abiotic stress factors affecting agricultural production worldwide, and salt stress has significant impacts on plant growth and development. Under salt treatment, seed germination, root length, plant height and fruit development are significantly inhibited [1]. Salt stress can decrease cotton yield by up to 50–67% [2]. The availability of salt-tolerant varieties would expand the area of cotton production by promoting the synchronous growth of food crops and cotton. Therefore, screening and utilizing excellent salt–alkali-tolerant germplasm resources and salt-tolerant genes play important roles in the transformation and utilization of saline–alkali land and improving the level of agricultural production in saline–alkali land. The damage to cotton caused by salt stress is mainly related to the effects of salt ions on the structure and function of young organs or cell membranes during the developmental transition period, which inhibits the growth of cotton seedlings, affects the growth process, reduces the number of total fruit nodes and reduces yield and quality [3,4,5].

Salt tolerance is a complex quantitative trait, which is controlled by multiple genes and involves a variety of physiological and biochemical metabolic pathways in cotton [6]. Quantitative trait locus (QTL) mapping of salt-related traits has become an effective approach. Compared to traditional QTL mapping, GWAS uses genome-wide SNPs as molecular markers to dissect the genetic basis of complex traits [7]. High-throughput genotyping platforms play an important role in plant genome research. GWAS was conducted for salt tolerance at the seedling stage using 419 upland cotton accessions and 17,264 SNPs [8]. Similarly, Cai et al. (2017) constructed a high-density 80K SNP cotton chip that contained 77,774 SNP sites, among which 352 cotton materials were analysed, and 76.51% of the sites were polymorphic [9]. The chip was utilized to perform a GWAS analysis of 288 upland cotton materials and found that eight SNPs were significantly associated with three salt tolerance traits. A GWAS was performed to identify marker-trait associations and 713 upland cotton accessions were phenotyped under salt stress and genotyped by the Illumina Infinium CottonSNP63K array [10]. Huang et al. (2017) used the US 63K chip to perform GWAS analysis on 503 upland cotton resources (63K) and identified 324 SNPs and 160 QTLs related to 16 agronomic traits, of which 38 related areas controlled 2 or more traits [11]. Paterson et al. (2012) used a population of upland cotton (440 materials) and a population of sea island cotton (219 materials) and the genotyping-by-sequencing (GBS) method to develop 10,129 SNP markers, and they obtained monomer domains in the whole gene range through analysis [12]. Abdelraheem et al. (2020) studied tolerant and sensitive alleles recombined for tolerance to the abiotic stress during the intermating process for population development: 23 QTL were detected for salt tolerance including 12 and 11 QTL for plant height and dry shoot weight, respectively [13]. Many QTLs were mapped on A05, A08, A09, A11, D01, D08, D09, D10 and D12 for salt tolerance in tetraploid cotton. Xu et al. (2021) found a locus on chromosome A13 and D08 associated with relative plant height (RPH), and loci on A07 for relative shoot fresh matter weight (RSFW); A08 and A13 for relative shoot dry matter weight (RSDW) were expressed in both environments and the increase in GH_A13G0171-silenced plants’ salt tolerance under salt stress indicated its negative function in regulating the salt stress response [6].

In recent years, with increasing rainwater and serious secondary salinization, it has been crucial to study the salt tolerance traits of cotton at seedling stage. Cultivation of salt-tolerant crop varieties is an important way to achieve sustainable agricultural development in the future. Similar to other crops, excessive salts in the soil severely affect germination and seedling, particularly at sensitive stages in the life of a plant [14,15,16]. However, the salt tolerance of plants is a very complicated process. In this study, 149 elite *Gossypium hirsutum* cultivar accessions (lines) were used as materials, and 70K SNP chips were used to screen SNP loci and perform genome-wide association analysis on the traits related to salt tolerance at the seedling stage to find significant association sites related to salt tolerance. This study provides a reference and basis for further theoretical studies, such as the isolation of related genes and molecular marker-assisted selection of cotton salt tolerance.

## 2. Results

### 2.1. Genetic Diversity Analysis and Population Structure

#### 2.1.1. Group Structure

It is important for GWAS analysis to control the effect of population structure because population stratification could eliminate spurious associations between genotypes and phenotypes [17,18]. In total, 18,430 highly consistent SNP sites were obtained. PCA found that 149 individuals of upland cotton could be divided into two subgroups, including subgroup 1 (marked in red) consisting of 78 materials, and subgroup 2 consisting of 71 individuals, with PC1 9.25% and PC2 5.2%, respectively. Based on the analysis of the phylogenetic tree constructed from the SNP data, the 149 cotton plant accessions could be divided into two subgroups. The results are quite consistent with the grouping structure of PCA (Figure 1B). The hybrid model was used in ADMIXTURE V250 software. First, the number of subgroups (K) was set to 1–20, and each K value was set to three repetitions. Assuming that each site was independent, the Markov chain Monte Carlo (MCMC), at the beginning of the non-count iteration (length of burn-in period), was set to 10,000 times, and then the MCMC, after no-count iterations, was set to 1,000,000 times. The optimal K value was selected according to the principle of maximum likelihood value to determine the number of subgroups and the group structure. The cross-validation error (CV error) was calculated under different K values (2–20). In this experiment, using the Q value calculation and Structure software, when K was equal to 18, the CV error value was the smallest (Figure 1C). Taking the corresponding Q-matrix at k = 18 as the covariate could reasonably eliminate spurious association effects and improve the GWAS accuracy. The cotton population showed a certain distribution gap, but most varieties were clustered together, which corresponded to the actuality that the early introduction of Chinese land cotton varieties were mainly from the United States and the Soviet Union, and breeding used the introduced materials as the parents. The results showed that the division of the subgroups was significantly correlated with the material source, indicating that the genetic background of the resources was relatively homogeneous. For subgroups, the results of population structure analysis were in line with the evolutionary trends of the genetic background during breeding.

#### 2.1.2. Material Heterozygosity

The individual heterozygosity analysis found that 95% of the cotton accessions were less than 30% heterozygous, and 80% of the individual materials were less than 5% heterozygous (Figure 2). Possible misalignment caused by homologous exchange (HEs) was prevalent in heterologous tetraploid crops, but, in cotton, HEs remained at a low level according to previous studies and had little effect on the results [19].

#### 2.1.3. Kinship Distribution

Figure 3 shows that the 149 cotton varieties can be divided into two subgroups. Among the 149 varieties, the genetic relationships between most varieties were weak (the yellow parts in the figure), and the genetic relationships between a few materials were very close (the dark red parts in the picture).

#### 2.1.4. Analysis of Linkage Disequilibrium

The LD distance decreases as the physical position of the SNP on the chromosome increases. The analysis found that the LD distance of 149 samples was 432 kb (R square = 0.5) (Figure 4). Slightly higher than the previous study [20] where Chinese upland cotton material was 296 kb. This result further shows that the genetic diversity of the selected material is reduced. The genetic differentiation history of cotton collected in China is relatively low. This is related to the tended homogenization of Chinese cotton breeding.

### 2.2. Phenotypic Statistical Analysis

In the best linear unbiased prediction (BLUP) for salt tolerance index traits, a total of five salt tolerance index phenotypic traits, 3d Germination potential index, 3d Radicle length index, 7d Radicle length index, 7d Radicle weight index, 7d Germination rate index, were identified. Figure 5 shows that the phenotypic distributions of the five traits were all normally distributed (*p*-value > 0.05), indicating that these traits are all typical quantitative traits and are controlled by minor-effect polygenes. Using R language to calculate the Pearson correlation coefficients between traits, it was found that the correlations between different traits were low (Table 1).

### 2.3. Association Analysis of Salt Tolerance Traits

Based on the identification results of the morphological, physiological, biochemical and yield traits of the specific germplasms of upland cotton under saline–alkali stress, an association analysis of the salt tolerance traits was carried out, and the favorable alleles related to the salt tolerance of the specific germplasm of upland cotton were located. A total of 27 SNP sites related to salt tolerance traits were detected (Table 2): 3 SNP sites related to 7d Radicle length; 10 SNP sites related to 7d Radicle length drop rate; 3 SNP sites related to 7d Germination rate; 3 SNP sites related to 7d Germination weight; 8 SNP sites related to Relative germination rate. The marker loci were scattered on six cotton chromosomes, A01, D01, D05, D08, D11 and D13, without clustering, and six QTLs related to salt tolerance traits were located on different chromosomes (Table 2). The research and development of SNP markers and QTL sites closely related to salt tolerance traits can be applied to the molecular marker-assisted selection of cotton salt tolerance.

### 2.4. Association Analysis of Salt Tolerance Index Traits among Cotton Accessions

The results of the GWAS under the optimal model of the salt tolerance index traits using BLUP were counted and explained, and the results are shown in Table 2. A total of 15 significant SNP-trait associations were detected (−log10(*p*) > 3) (Table 3). It was also found that among these 15 traits, only 4 traits had significant SNPs, while the 7d Radicle weight index did not have a significant locus. This may be because this trait is more complicated and controlled by multiple minor QTLs (Figure 6). According to the Bonferroni correction principle, −log10(*p*) > 3.97 (*p* = 1/n, n is the SNP numbers in this study) should be the threshold, but the Bonferroni correction is too stringent and no significant SNPs for two traits could be identified with this threshold. To obtain more associated SNPs, the significantly associated SNP markers with salt-tolerant-related traits were identified according to −log10(*p*) > 3.0 [21,22].

### 2.5. Candidate Gene Screening

To investigate the expression pattern of these genes during the seedling stage under salt stress tolerance, and to further screen the possible candidate genes involved in the salt response, these genes were analyzed using the expression level of the seedlings at 1, 3, 6 and 12 h under 400 mM salt concentration. The public transcriptome data sets were retrieved from ccNET (https://structralbiology.cau.edu.cn.gossypium (accessed on 22 February 2020)) [23]. As a result, five notable SNP sites on chromosome A01 and two notable SNP sites on chromosome D01 associated with the 7d Radicle length drop rate were detected, three genes, *Gh_D01G0943*, *Gh_D01G0944* and *Gh_D01G0945*, were found at 500 kb upstream and 500 kb downstream of the two relevant SNP sites in D01; ten genes, *Gh_A01G0905*, *Gh_A01G0906*, *Gh_A01G0907*, *Gh_A01G0908*, *Gh_A01G0909*, *Gh_A01G0910*, *Gh_A01G0911*, *Gh_A01G0912*, *Gh_A01G0913* and *Gh_A01G0914*, were found at 500 kb upstream and 500 kb downstream of the five relevant SNP sites in A01. At the same time, six identical associated SNP loci were detected in the Relative germination rate and 7d Germination rate index, all of which were on the D08 chromosome. Ten genes were detected in the upstream and downstream 500 kb of the associated SNP loci: *Gh_D08G1305*, *Gh_D08G1306*, *Gh_D08G1307*, *Gh_D08G1308*, *Gh_D08G1309*, *Gh_D08G1310*, *Gh_D08G1311*, *Gh_D08G1312*, *Gh_D08G1313* and *Gh_D08G1314* (Figure 7). These genes were aligned to the transcriptome data, selected for differentially expressed genes, and, finally, we identified 6 candidate genes associated with plant salt stress by gene functional annotation with previous findings (Table 4). The transcriptome results showed that *Gh_D01G0945, Gh_A01G0906* and *Gh_D08G1308* expression was down-regulated of salt stress; *Gh_A01G0908* and *Gh_D08G1309* expression was up-regulated of salt stress. *Gh_D01G0943* expression reached their peak at 3 h of salt stress (Figure 8).

### 2.6. Alignment of Salt Resistance-Related Genes and Arabidopsis Homologous Sequences

The comparison of *Gh_**D**01G0**943* with Tair3 showed that the homologous gene in *Arabidopsis thaliana* is *AT1G75680*, with 76% homology, and the gene is an *GH9B7* glycosyl hydrolase 9B7. The comparison of *Gh_D01G0945* with Tair3 showed that the homologous gene in *Arabidopsis thaliana* is *AT1G77210*, with 75% homology, which encodes *STP14* sugar transporter 14. The comparison of *Gh_**A**01G0**906* with Tair3 showed that the homologous gene in *Arabidopsis thaliana* is *AT**2**G**21220*, with 78% homology, and the gene is an SAUR-like auxin-responsive protein family. The comparison of *Gh_**A**01G0**908* with Tair3 showed that the homologous gene in *Arabidopsis thaliana* is *AT**1**G**19850*, with 83% homology, and the gene is an *MP* Transcriptional factor B3 family protein/auxin-responsive factor AUX/IAA-like protein. The comparison of *Gh_**D**0**8**G**1308* with Tair3 showed that the homologous gene in *Arabidopsis thaliana* is *AT**5**G**18010*, with 83% homology, and the gene is an *SAUR19* SAUR-like auxin-responsive protein family. The comparison of *Gh_**D**0**8**G**1309* with Tair3 showed that the homologous gene in *Arabidopsis thaliana* is *AT**4**G**02280*, with 75% homology, and the gene is an *SUS3* sucrose synthase 3 (Figure 9, Table 5). Both *Gh_A01G0906* and *Gh_D08G1308 Arabidopsis thaliana* orthologs belong to SAUR-like auxin-responsive protein family, so both genes may have similar gene functions, the transcriptomic results showed similar expression patterns of both genes.

## 3. Discussion

### 3.1. Target Gene Identification Based on GWAS

With the efficient development of genotyping technology, SNP markers have the advantages of wide distribution, high throughput, low cost and high accuracy. Genome-wide association analysis based on SNP genetic markers has become the first choice for analyzing the complex traits of humans, animals and plants [24]. Some significant SNP markers in linkage disequilibrium can show a higher degree of linkage disequilibrium than those SNPs that actually cause phenotypic variation [25,26].

Sun et al. (2018) identified 31 SSRs and 8 SNPs associated with salt tolerance based on relative seed germination rate under seven environments, using 503 upland cotton accessions and 179 SSRs and 11,975 array-derived SNPs [27]. Du et al. (2016) performed association analysis of 304 upland cotton cultivars and identified 95 significant associations for 10 salt tolerance-related traits at the germination and seedling stages [28]. Jia et al. (2014) identified three simple sequence repeat (SSR) markers significantly associated with the relative survival rate under salt stress through association mapping methods using 323 *G. hirsutum* germplasms. Two haplotypes related to fibre length and fibre strength were identified on chromosomes At07 and Dt11 [29]. Reddy et al. (2017) used GBS (genotyping-by-sequencing) SNP typing technology to develop 10,129 polymorphic SNP markers from upland cotton and sea island cotton based on SNP markers and linkage disequilibrium LD from upland cotton and sea island cotton. A total of 142 and 282 blocks were excavated from sea island cotton [30].

In this study, Illumina Cotton SNP 70K was used to develop 18,430 SNP markers in the whole genome. On this basis, whole-genome association analysis was used to associate excellent sites related to salt tolerance traits and the salt tolerance index. The results of a total of 27 SNP sites related to salt tolerance traits were detected. The GWAS under the optimal model of the salt tolerance index traits using BLUP were counted and explained, a total of 15 significant SNP-trait associations were detected.

### 3.2. Functional Analysis of Candidate Genes

Candidate genes are a class of genes whose expression on the chromosome is not clear. They are involved in the phenotypic expression of organisms, and association analysis suggests that they are related to a certain part of the genome. Such genes may be structural genes, regulatory genes or affect the expression of traits in biochemical metabolic pathways. The functional insufficiency of the candidate gene is known, and whether it is related to salt resistance has been verified. According to the screening, functional annotations can be assigned, or *Arabidopsis* homologous genes can be found from the gene information. This method has been previously reported to target genes that are clearly related to salt tolerance. GWAS analysis is a fast and powerful method to mine regulatory genes through crop indicators. A number of genes conferring salt tolerance such as MKK [31], ZFP [32], NAC [33], ERF [34], DREB [32], GhMT3a [35], MPK [36] and tonoplast Na+/H+ antiporter [37] have been identified in cotton. Sun et al. (2018) reported that, among a total of 223 genes within a salt tolerance QTL interval on D01 (37,771–1,942,912), four candidate genes (*GhPIP3A*, *GhSAG29*, *GhTZF4* and *GhTZF4a*) showed a differential expression between sensitive and tolerant accessions under salt stress [10]. In this study, the comparison between our GWAS under the optimal model of the salt tolerance index traits for BLUP and previous reports showed that most of the 15 associations were novel reported loci. Abdelraheem et al. (2017, 2018) reported that each chromosome in each of the six pairs of homologous chromosomes (i.e., A01/D01, A03/D03, A08/D08, A09/D09, A12/D12 and A13/D13) had at least one ST (salt stress) QTL > 470,000 SNPs surveyed [38,39]. A total of 16 and 27 QTLs were identified for dry shoot weight and plant height, respectively, under both water-limited and saline environments, while 11 QTLs were found commonly linked with tolerance to drought and saline environments. Five striking SNP sites on chromosome A01, two striking SNP sites on chromosome D01 associated and six striking SNP sites on chromosome D08 were detected. The *Gh_D01G0943*, *Gh_D01G0945*, *Gh_A01G0906*, *Gh_A01G0908*, *Gh_D08G1308* and *Gh_D08G1309* gene associated with plant salt stress was detected at the associated SNP locus.

Sucrose synthase 3 (*SUS**3*) catalyzes the reversible conversion of sucrose and a nucleoside diphosphate into the corresponding nucleoside diphosphate-glucose and fructose; a member of the glycosyltransferase family of enzymes, sucrose synthase 3 is ubiquitous in the plant kingdom and catalyzes in vivo and in vitro the synthesis and cleavage of sucrose. Sucrose synthase is one of the key enzymes in the plant carbohydrate metabolism and regulates the assignment of sucrose, a kind of product of photosynthesis, into a variety of plant metabolic processes. Sucrose synthase also plays pivotal roles during plant growth and development [40,41]; SUS is important for metabolite homeostasis and the timing of seed development and is a key enzyme of carbon metabolism in the heterotrophic tissues of plants [42,43]. SUS can transport sucrose into a variety of pathways, the most important of which provides a precursor substance (UDP-glucose) for the biosynthesis of cell wall polymers and starch [44]. In addition, SUS plays an important role in the process of growth, development and the metabolism of sink organs, and also plays an important role in the adaptation of plants to abiotic stress, such as hypoxia and cold; under hypoxia, the activity of SUS increased. *SUS3* is induced in various organs under dehydration conditions including leaves deprived of water or submitted to osmotic stress as well as late-maturing seeds [45]. 

Sugar transporter 14 (*STP14*), as a transport protein, is a galactose transporter expressed in both source and sink tissues with the highest levels in the endosperm; it affects galactose transmembrane transporter activity, carbohydrate transmembrane transporter activity and sugar–hydrogen symporter activity [46]. STP are proton-coupled symporters responsible for the uptake of glucose from the apoplast into plant cells. They are integral to organ development in symplastically isolated tissues such as seed, pollen and fruit. Additionally, STPs play a vital role in plant responses to stressors such as dehydration and prevalent fungal infections such as rust and mildew [47]. STPs play a role in senescence and programmed cell death, and participate in the recycling of sugars derived from cell wall degradation [46,48,49].

Glycosyl hydrolase 9B7 (*GH9B7*) is a kind of enzyme that hydrolyzes glycosidic bonds and plays an important role in the hydrolysis and synthesis of biological sugar and sugar conjugates. When the enzyme catalyzes the glycosidic reaction, if the oxygen atom of water molecule attacks the anomeric carbon on the receptor glucose, it will be hydrolyzed, but if the oxygen atom on the hydroxyl group of glucose attacks the anomeric carbon on the receptor glucose, it will be transglycosylated [50]. *GH9B7* belong to glucosidase and their main function is to hydrolyze the glucoside bond, releasing glucose as a product. They are an indispensable class of enzymes in the glucose metabolism pathway of living organisms, involved in the carbohydrate metabolic process [51]. The process of N-glycosylation involves the participation of various enzymes, mainly glycoacyltransferases, including the transfer of active donors (usually NDP-sugar) to molecules of recipient substances such as sugars, proteins and lipids; the latter catalytic activity is to trim various glycochains, which together complete N-glycosylation. According to the current research results, it can be determined that the N-glycosylation modification of the protein plays an important role in the processes of protein folding and transportation [52]. 

Transcriptional factor B3 family protein/auxin-responsive factor AUX/IAA-like protein (*MP*), is an auxin-responsive transcription factor that is required for primary root formation and vascular development [53]. It plays a critical role in *Arabidopsis* embryonic root initiation, *MP* transcriptionally initiates the ground tissue lineage and acts upstream of the regulatory network that controls ground tissue patterning and maintenance [54]. In the shoot, cell polarity patterns follow MP expression, which in turn follows auxin distribution patterns [55]. Signaling through *MP/AUXIN RESPONSE FACTOR 5* is necessary for the formation of shoots from *Arabidopsis calli* [56]. Aux/IAA auxin perception mediates rapid cell wall acidification and growth of *Arabidopsis hypocotyls* [57]. AUX/IAA is a transcriptional repressor that has proved to play a very vital role in the auxin signaling pathway [58].

The plant hormone auxin controls numerous aspects of plant growth and development by regulating the expression of hundreds of genes. SMALL AUXIN UP RNA (*SAUR*) genes comprise the largest family of auxin-responsive genes; the *SAUR19-24* subfamily of auxin-induced *SAUR* genes promotes cell expansion [59]. SAUR proteins provide a mechanistic link between auxin and plasma membrane H+-ATPases (*PM H+-ATPases*) in *Arabidopsis thaliana*. Plants overexpressing stabilized *SAUR19* fusion proteins exhibit increased *PM H+-ATPase* activity, and the increased growth phenotypes conferred by *SAUR19* overexpression are dependent upon normal *PM H+-ATPase* function. *SAUR19* stimulates *PM H+-ATPase* activity by promoting phosphorylation of the C-terminal autoinhibitory domain. *SAUR19*, as well as additional SAUR proteins, interacts with the *PP2C-D* subfamily of type 2C protein phosphatases. These phosphatases are inhibited upon SAUR binding, act antagonistically to SAURs in vivo, can physically interact with *PM H+-ATPases* and negatively regulate *PM H+-ATPase* activity [60]. SAURs play a central role in auxin-induced plant growth, but can also act independently of auxin, on tissue specifically regulated by various other hormone pathways and transcription factors [61]. 

Auxin functions, at least in part, by regulating a set of early auxin response genes: Aux/IAAs and SAURs [62]. Auxin is perceived by receptors including TRANSPORT INHIBITOR RESPONSE 1 (*TIR1*) and the closely related AUXIN SIGNALLING F-BOX (*AFB*) F-box proteins, which recruit Aux/IAA repressors to the SCFTIR1/AFB complex for ubiquitination and proteasome-mediated degradation, releasing the inhibition of AUXIN RESPONSE FACTORS (*ARFs*), and eventually activating auxin-induced gene expression [63]. The root growth inhibited by salt stress was related to the decrease in auxin accumulation. The position of auxin transporter *AUX1* is changed due to salt stress, so auxin transport may be related to decreased accumulation of auxin in the roots [64]. The accumulation of carbohydrates such as sucrose plays an important role in alleviating stress damage, including osmotic protection, carbon source storage and *ROS* removal. ASISH et al. (2004) showed that the intracellular reducing sugar (sucrose and fructan) levels of different species of plants are increased under salt stress [65]. In the process of salt stress response, the mechanisms or strategies that control the metabolism, transportation and balance of molecules, hormone metabolism, antioxidant metabolism and signal transduction mechanisms play a vital role in the process of plant adaptation to salt environment.

## 4. Conclusions

A total of 18,430 polymorphic SNP markers were developed and screened from natural populations using gene chip technology. These SNP markers were used to analyse the structure of the population to obtain the Q-matrix, and then the salt tolerance traits and salt tolerance index data were combined to conduct a genome-wide association analysis. The natural population can be divided into two subgroups. The genetic relationship between the cotton cultivars was weak; indicating that the breed inherited diversity is decreasing. The salt tolerance traits were associated with 27 significant SNP sites, and the salt tolerance index was associated with 15 significant SNP sites. The significant SNP sites were further analysed, salt tolerance-related *Gh_D01G0943*, *Gh_D01G0945*, *Gh_A01G0906*, *Gh_A01G0908*, *Gh_D08G1308* and *Gh_D08G1309* were detected in the spot data. The homologous sequences were compared with *Arabidopsis thaliana* to obtain the homologous genes *AT1G75680*, *AT1G77210*, *AT2G21220*, *AT1G19850*, *AT5G18010* and *AT4G02280*. Analysis of the functions of these six genes revealed that the *Arabidopsis thaliana* homologous sequence encodes the glycosyl hydrolase 9B7, sugar transporter 14, sucrose synthase 3, SAUR-like auxin-responsive protein family and Transcriptional factor B3 family protein/auxin-responsive factor AUX/IAA-like protein. The sucrose-generating metabolic system, transmembrane transport system and regulation of the auxin response have high activity in the salt tolerance reaction of cotton, so the stability of the structure and function of the protective membrane and macromolecular matter are generated to maintain the cellular osmotic pressure balance and are the key to the salt tolerance of cotton. This study further analysed the functions and expression patterns of cotton salt-tolerant genes and even has certain reference value for analyzing the mechanism of cotton salt tolerance.

## 5. Materials and Methods

### 5.1. Test Materials

We sampled 149 modern *G. hirsutum* cultivars collected from the Chinese national medium-term cotton gene bank at the Institute of Cotton Research (ICR) of the Chinese Academy of Agricultural Sciences (CAAS) (Table 6).

### 5.2. DNA Extraction and Genotyping

All cotton seeds were grown in a soil mixture in a fully automated greenhouse under a 12-h light/12-h dark cycle at 28 °C. Total genomic DNA was extracted from 5-day-old seedlings germinated from five well-developed seeds of each accession using a Qiagen D Neasy plant mini kit (Qiagen, CA, USA) following the protocol provided by the manufacturer. Genotyping was conducted at the CapitalBio Technology Platform in China using the Illumina Cotton SNP 70k Beadchip (Illumina, Inc., San Diego, CA, USA). All genotype SNP calls were extracted from the raw data using GenomeStudio (Illumina). A QTL was declared when four significant SNPs at 1 × 10^−3^ were detected within a 6 Mb region, and a QTL cluster was declared when multiple QTL were overlapped or were within a 15–40 Mb region [66,67,68]. The reference genome was:

*Gossypium hirsutum* (AD1) ‘TM-1′ genome NAU-NBI v1.1 a1.1.

### 5.3. Molecular Genetic Diversity and Phylogenetic Analyses

PHYLIP (http://evolution.genetics.washington.edu/phylip.html (accessed on 15 February 2020)) was used to calculate the genetic distance matrix of the sample, Notepad++ software was used to adjust the genetic distance matrix file into a suitable format. After generating the tree file, iTOL (https://itol.embl.de/ (accessed on 18 February 2020)) was used to draw the NJ tree diagram. Software was used to construct based on the neighbor-joining method and the p-distance model and bootstrapping was repeated 1000 times. Principal component analysis (PCA) was performed on cotton population materials using GCTA 1.93 software [69] using the detected SNPs. Then, R software was used to calculate the vector of each principal component and draw the PCA scatter plot. The SNP data of 149 experimental materials were detected by Illumina Cotton SNP 70K and filtered according to the minor allele frequency (MAF: 0.05) and site integrity (INT: 0.1). The clean reads were anchored to the cotton reference genome using Burrows-Wheeler Aligner (BWA). The SAM tools software was used to convert alignment files to BAM files. After 63,058 probe sequences were blast aligned with the genome, the optimal result screened out was the position of the SNP on the reference genome. SnpEff 4.0 [70] software was used to obtain the locations of the variable sites (intergenic zones, gene zones or CDS zones) in the reference genome and the effects of the variations (synonymous mutations, nonsynonymous mutations, etc.).

### 5.4. Population Structure and Kinship Analysis

ADMIXTURE V250 [71] software was used to analyze the group structure of the research materials. For the research group, the number of subgroups (K value) was preset to 1–20 for clustering, the clustering results were cross-validated, and the optimal number of clusters was determined according to the lowest cross-validation error rate. SPAGeDi 1.3 [72] software was used to estimate the relative kinship between two individuals in a natural population. The kinship itself is the relative value that defines the genetic similarity between two specific materials and the genetic similarity between any materials. Therefore, when the kinship value between the two materials is less than 0, it is directly defined as 0. Five independent runs were performed; the number of populations (K) was set from 1 to 20; the burn-in time and Markov chain Monte Carlo replication numbers was set to 10,000. The optimal K value was determined by comparing the LnP (D) and Δk based on the rate of change in LnP (D) [73]. A Q-matrix produced by STRUCTURE listed the estimated membership coefficients in a cluster for the subsequent association analysis.

### 5.5. Linkage Disequilibrium Analysis

On the same chromosome, the linkage disequilibrium between two SNPs within a certain distance can be calculated (such as 1000 kb), and the linkage disequilibrium strength is represented by r^2^. The closer r^2^ is to 1, the stronger the strength of linkage disequilibrium. The SNP spacing is fit to r^2^, and a graph can be drawn to represent the variation of r^2^ with distance. Generally, the closer the SNP spacing is, the larger r^2^ is, and the farther the SNP spacing is, the smaller r^2^ is. The distance travelled when the maximum r^2^ value drops to half is used as the LD decay distance (LDD) of linkage disequilibrium. The longer the LDD is, the smaller the probability of recombination within the same physical distance; the shorter the LDD is, the greater the probability of recombination within the same physical distance. Plink2 [74] software was used for LD analysis.

### 5.6. Association Analysis of Salt Tolerance Traits

The TASSEL5.0 (http://www.maizegenetics.net/tassel (accessed on 26 February 2020)) software package, EMMAX (http://genetics.cs.ucla.edu/emmax/ (accessed on 26 February 2020)) software package and FaST-LMM0.2.19 (https://www.microsoft.com/en-us/download/confirmation.aspx?id=52588 (accessed on 22 February 2020)) software package were employed to construct association tests of salt tolerance-related traits. Through a certain amount of population SNP marker data, combined with population structure and target trait phenotype data, the target region or site associated with the target trait can be located.

### 5.7. Salt Stress Conditions and Salt-Tolerant Trait Collection

The salt tolerance test during the germination period used double-layer filter paper rolls to stand the plant upright. Two pieces of filter paper each 20 cm in length and width were cut, and one piece of filter paper was spread on the test bench with a sprayer containing NaCl solution. The filter paper was soaked, and 15 seeds were placed 2 cm down from the top of the filter paper. The filter paper was then placed vertically into the culture box. Approximately 30 rolled filter papers were placed in each culture box. The culture box was then placed at 28 °C, and the photoperiod was 10 h/14 h (L/D), with heat preservation and culture in a constant temperature light incubator. The germination potential of seeds and the length of each seed were measured on the 3rd day, and the germination rate, Radicle length and stem fresh weight of the seeds were measured on the 7th day. This process was repeated 3 times. The treatment concentrations of NaCl solution were 0 NaCl (CK) and 150.0 mmol/L NaCl (Table 7) [75,76,77,78].

The calculation formula analyzes the relative values of the salt stress environment and the control conversion. The germination standard is that the radicle is half the length of the seed. Germination weight is the weight of all biological materials after germination.

Relative germination potential % = germination potential of treated seeds/germination potential of control seeds × 100%.

Relative germination rate % = germination rate of treated seeds/germination rate of control seeds × 100%.

Decrease rate % = (treatment traits − control traits)/control traits × 100%.

Salt tolerance index:$$\mathrm{SI}=\frac{X_{d}}{\bar{X}}\times \frac{X_{d}}{X_{w}}$$

Note: X_d_ and X_w_ are the measured values of a certain index of each material under salt stress conditions and contrast conditions, respectively, and $\bar{X}$ is the average value of this index under salt stress conditions.

Germination potential:$$\mathrm{GP}=\frac{M_{1}}{M}$$

Note: M_1_: Number of normal germinating grains within days of germination potential; M: Number of seeds to be tested.

Statistical analysis of the phenotype of salt tolerance-related traits was performed by SPSS. SAS software was used to perform the best linear unbiased prediction (BLUP) for salt tolerance traits; the parameter is the default value. Software was used to perform correlation analysis for each trait based on the model of mlm, glm, cmlm, emmax and fastlmm, and the result of the structure was used as a fixed effect. Due to the small number of environments and the existence of certain false positives, the CMLM model can reduce the false positives as much as possible, so the method of CMLM is adopted. The CMLMs were performed by simultaneously accounting for multiple levels of Q-matrix and K-matrix according to the methods described [79]. Among them, the mixed linear model formula of TASSEL software is as follows:y=Xα+Qβ+Kμ+e

Note: SPAGeDi 1.3 [72] software was used to calculate the genetic relationship K between samples. The general linear model uses Q population structure information, while the mixed linear model uses Q + K, which is the population structure and genetic relationship information. X is the genotype and Y is the phenotype. In the end, an association result can be obtained for each SNP site.

Salt stress cotton transcriptional group data download: Sequencing of allotetraploid cotton (*Gossypium hirsutum* L. acc. TM-1) provides a resource for fibre improvement. Nat Biotechnol, 2015, doi:10.1038/nbt.3207 [23].

### 5.8. Prediction and Functional Annotation of Salt-Tolerant Candidate Genes

The independent significant SNP sites selected by the GWAS analysis results and LD calculations, plus or minus 500 kb upstream and downstream of the physical location of each SNP site as the candidate gene physical location query area, were identified by mapping the gene or *Arabidopsis* homologous gene and annotating information to narrow down the target candidate genes. NCBI, COTTONGEN, CNKI, Tair3 and other websites were used to annotate gene functions and compare homologous sequences.

## Figures and Tables

**Figure 1 plants-11-00097-f001:**
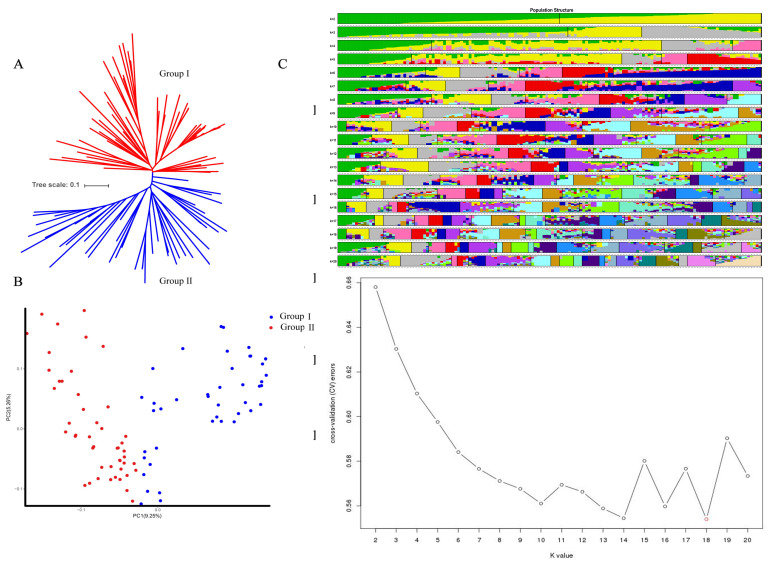
Analysis of groups based on structure. (**A**) NJ tree; (**B**) A scatter plot of principal component analysis (PCA); (**C**) Estimated population structure.

**Figure 2 plants-11-00097-f002:**
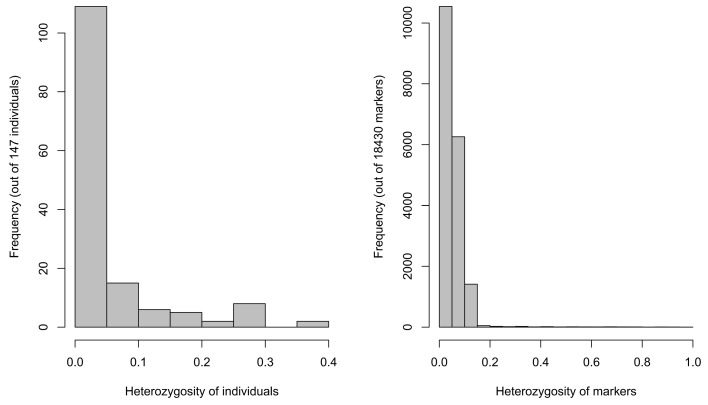
Frequency distribution of plant material heterozygosity and distribution of labelled heterozygosity.

**Figure 3 plants-11-00097-f003:**
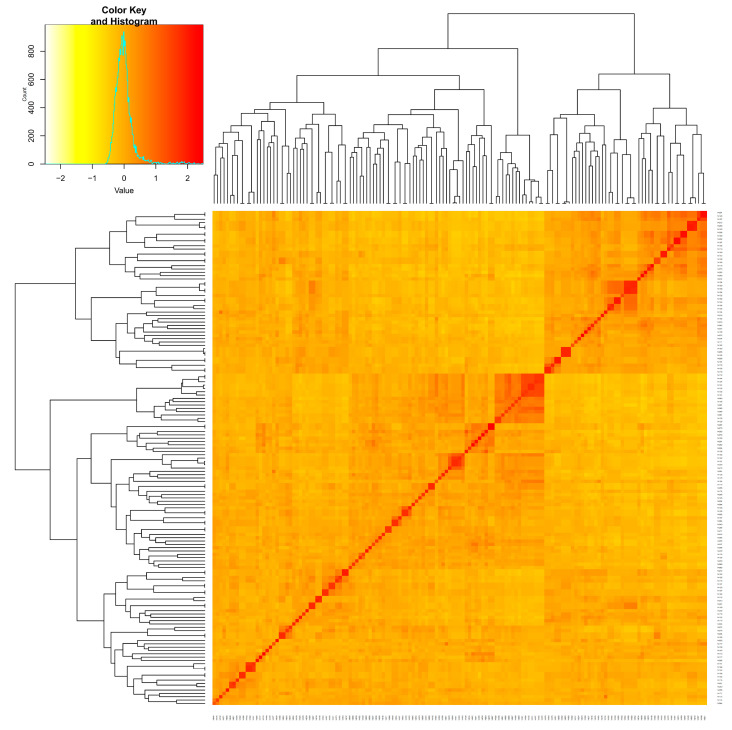
Heat map and cluster analysis based on the genetic relationship of 149 upland cotton varieties.

**Figure 4 plants-11-00097-f004:**
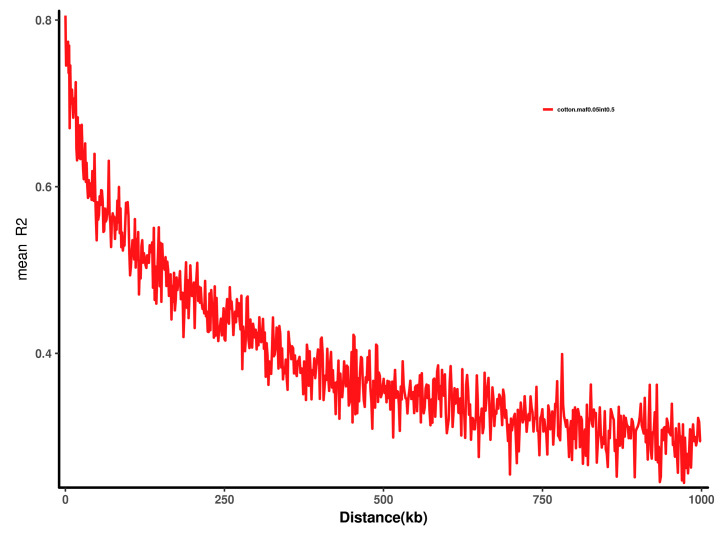
LD attenuation analysis of 149 cotton accessions.

**Figure 5 plants-11-00097-f005:**
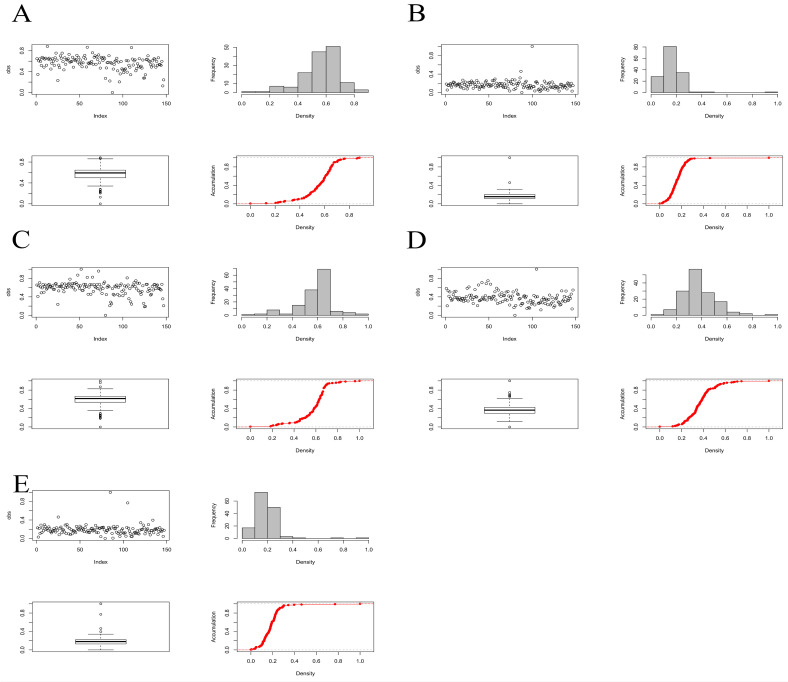
Distribution of traits in the three environments. (**A**) 3d Germination potential index BLUP analyse; (**B**) 3d Radicle length index BLUP analyse; (**C**) 7d Radicle length index BLUP analyse; (**D**) 7d Radicle weight index BLUP analyse; (**E**) 7d Germination rate index BLUP analyse.

**Figure 6 plants-11-00097-f006:**
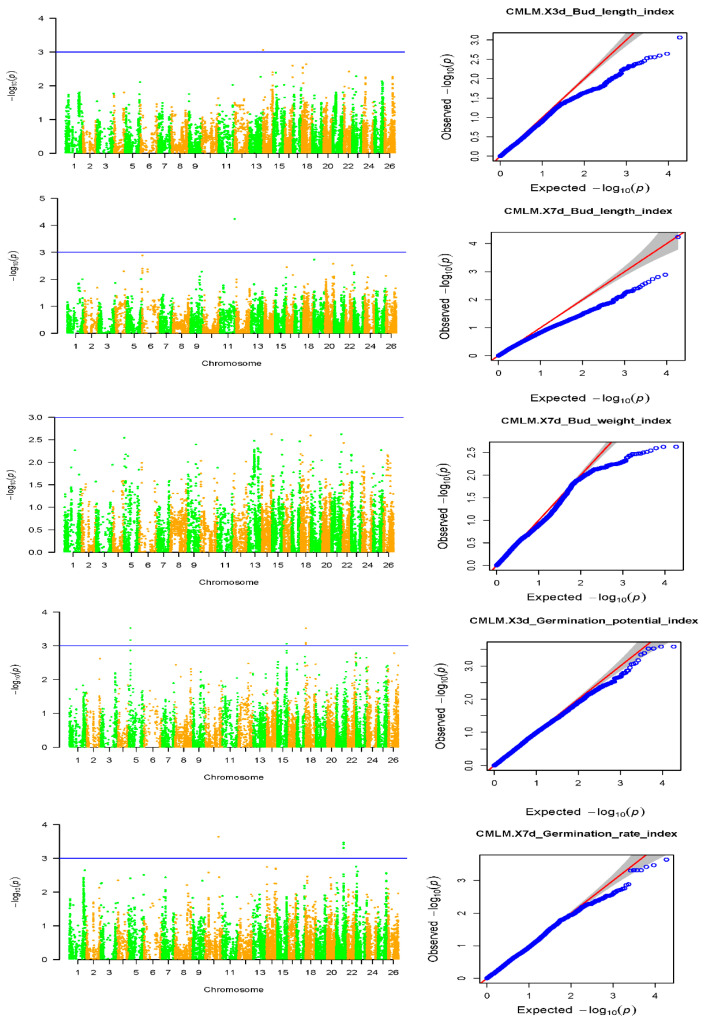
Manhattan and QQ charts of the 5 salt tolerance index traits. Manhattan graph: the abscissa represents the position of the chromosome, the ordinate represents the −log10(*p*) taking the negative logarithm based on 10, and the scattered dots (or lines) on the graph represent the corresponding data for each SNP site. The blue horizontal line is the threshold line. Scattered points (or lines) that exceed the threshold line are candidate sites. QQ chart: The abscissa represents the expected value, and the ordinate represents the observed value. The red line in the figure represents the 45° centre line, and the grey area is the 95% confidence interval of the scattered points in the figure.

**Figure 7 plants-11-00097-f007:**
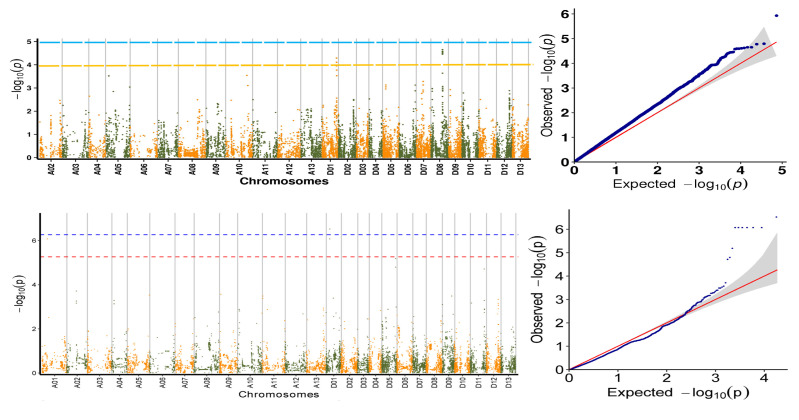
Screening of candidate genes for salt tolerance. Manhattan diagram: The abscissa represent chromosomal positions, the ordinate represents the *p*-value (−log10 (*p*)) with a negative logarithm at 10, and the scatter (or lines) on the figure represent the −log10 (*p*) corresponding to each SNP site. Blue horizontal line represents the value corresponding to 0.01/marker quantity, and red horizontal line represents the value corresponding to 0.1/marker quantity. A scatter (or line) above the threshold line is the candidate site. QQ chart: The abscissa represents the expected value, and the ordinate represents the observed value. The red line in the figure represents the 45° centre line, and the grey area is the 95% confidence interval of the scattered points in the figure.

**Figure 8 plants-11-00097-f008:**
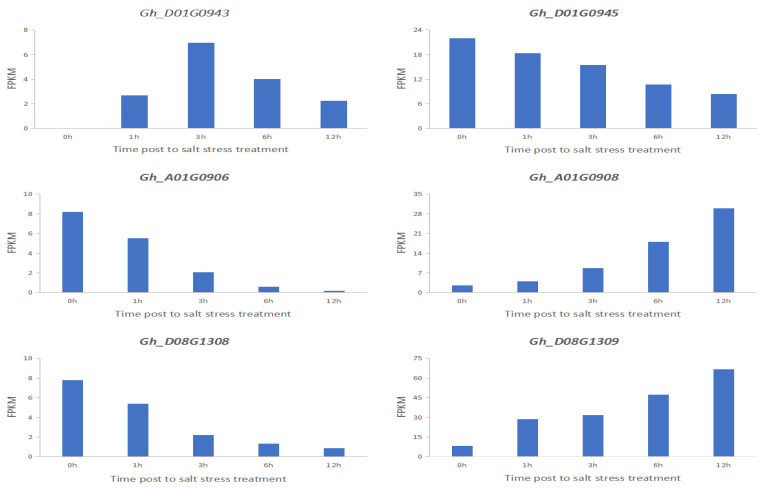
Transcriptome differentially expressed gene.

**Figure 9 plants-11-00097-f009:**
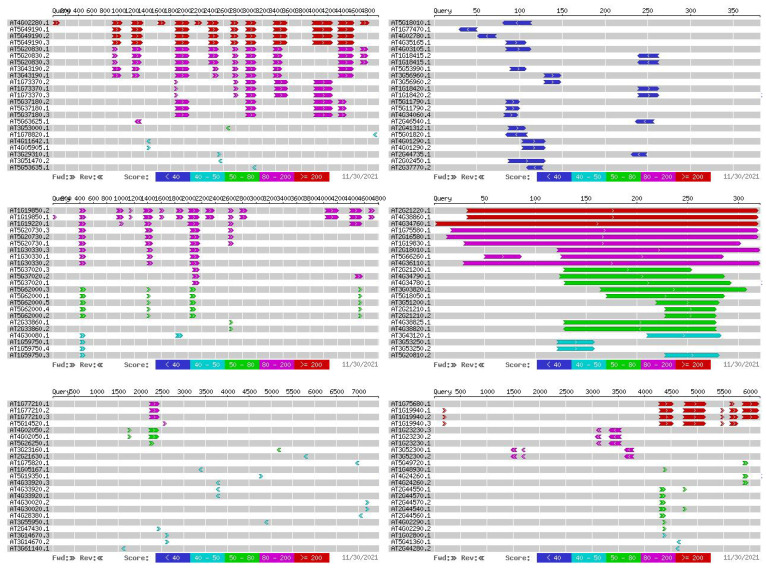
Gene sequence blast results.

**Table 1 plants-11-00097-t001:** Correlation analysis of salt tolerance index traits.

	3d Germination Potential Index	7d Germination Rate Index	7d Radicle Weight Index	3d Radicle Length Index	7d Radicle Length Index
3d Germination potential index	1	0.565096	0.330095	0.006608	0.178595
7d Germination rate index	0.565096	1	0.484635	0.000286	−0.03288
7d Radicle weight index	0.330095	0.484635	1	0.475708	0.192883
3d Radicle length index	0.006608	0.000286	0.475708	1	0.230305
7d Radicle length index	0.178595	−0.03288	0.192883	0.230305	1

**Table 2 plants-11-00097-t002:** SNP sites related to salt tolerance traits in cotton accessions.

Traits	Chromosomes	Position	*p*-Value	−log10(*p*)	Alleles
7d Radicle length	D08	64,320,143	0.000019	4.731944	T/G
	D13	35,877,478	0.000046	4.334531	C/T
	scaffold33030	93	0.000099	4.004747	A/C
7d Radicle length drop rate	D01	16,458,717	0.000000	6.588126	T/C
	A01	22,140,802	0.000001	6.1328	T/C
	A01	22,174,537	0.000001	6.1328	T/C
	A01	22,350,123	0.000001	6.1328	G/A
	A01	22,376,230	0.000001	6.1328	G/A
	A01	22,387,777	0.000001	6.1328	T/G
	D01	16,458,204	0.000001	6.1328	C/A
	D05	58,225,028	0.000006	5.226497	T/C
	D05	58,227,014	0.000015	4.83936	T/C
	D11	58,447,888	0.000017	4.762019	C/T
7d Germination rate	D05	12,154,352	0.000028	4.560614	C/A
	D05	12,155,655	0.000030	4.516439	C/T
	A05	12,181,224	0.000078	4.109102	G/A
7d Germination weight	D08	2,191,589	0.000054	4.264409	G/A
	D05	12,154,352	0.000056	4.255444	C/A
	D05	12,155,655	0.000092	4.035935	C/T
Relative germination rate	D08	43,495,093	0.000022	4.65032	A/G
	D08	43,541,365	0.000027	4.57704	A/G
	D08	43,557,843	0.000029	4.5365	T/C
	D08	43,483,056	0.000033	4.48071	A/G
	D08	43,501,088	0.000034	4.473257	C/T
	D08	43,479,511	0.000035	4.458785	C/T
	D01	54,316,248	0.000051	4.28993	T/C
	D01	54,289,162	0.000078	4.106987	C/A

**Table 3 plants-11-00097-t003:** SNP sites related to salt tolerance index traits in cotton accessions.

Traits	Chromosomes	Position	*p*-Value	−log10(*p*)	Alleles
7d Radicle length index	A11	85,527,572	0.000058	4.236572	T/C
7d Germination rate index	A10	79,275,413	0.000229	3.640165	C/T
	D08	43,479,511	0.000485	3.314258	C/T
	D08	43,483,056	0.000485	3.314258	A/G
	D08	43,495,093	0.00034	3.468521	A/G
	D08	43,501,088	0.000485	3.314258	C/T
	D08	43,541,365	0.000384	3.415669	A/G
	D08	43,557,843	0.000495	3.305395	T/C
3d Germination potential index	A05	12,472,578	0.000299	3.524329	A/G
	A05	12,473,346	0.000675	3.170696	A/C
	D02	56,360,435	0.000872	3.059484	A/G
	D05	3,861,440	0.000815	3.088842	T/C
	D05	3,870,101	0.000887	3.052076	A/G
	D05	3,873,693	0.000301	3.521434	G/A
3d Radicle length index	D01	883,613	0.000872	3.059484	C/T

**Table 4 plants-11-00097-t004:** Genes associated with salt stress.

Chromosomes	Gene	Position
D01	Gh_D01G0943	15,976,403–15,982,586
	Gh_D01G0945	16,034,346–16,041,810
A01	Gh_A01G0906	21,801,560–21,801,880
	Gh_A01G0908	22,204,069–22,208,876
D08	Gh_D08G1308	43,063,260–43,063,640
	Gh_D08G1309	43,067,294–43,072,264

**Table 5 plants-11-00097-t005:** Genes associated with salt stress.

Chromosomes	Gene	Arabidopsis Homology Gene	Homology Index
D01	Gh_D01G0943	AT1G75680	76%
	Gh_D01G0945	AT1G77210	75%
A01	Gh_A01G0906	AT2G21220	78%
	Gh_A01G0908	AT1G19850	83%
D08	Gh_D08G1308	AT5G18010	83%
	Gh_D08G1309	AT4G02280	75%

**Table 6 plants-11-00097-t006:** Sample number and name of cotton accessions/cultivars.

Number	Breed Name	Origin	Number	Breed Name	Origin	Number	Breed Name	Origin
1	Xinluzao No. 1	Inland Northwest	51	Xinluzhong No. 2	Inland Northwest	101	Coker310	Central Asia
2	Xinluzao No. 2	Inland Northwest	52	Xinluzhong No. 6	Inland Northwest	102	Dunhuang 77-116	Inland Northwest
3	Xinluzao No. 3	Inland Northwest	53	Xinluzhong No. 7	Inland Northwest	103	Shan 63-1	Yellow River Basin
4	Xinluzao No. 4	Inland Northwest	54	Xinluzhong No. 8	Inland Northwest	104	Shanmian No. 9	Yellow River Basin
5	Xinluzao No. 5	Inland Northwest	55	Xinluzhong No. 9	Inland Northwest	105	G164 2 20	Yellow River Basin
6	Xinluzao No. 7	Inland Northwest	56	Xinluzhong No. 10	Inland Northwest	106	Jimian No. 11	Yellow River Basin
7	Xinluzao No. 8	Inland Northwest	57	Xinluzhong No. 11	Inland Northwest	107	Lumian No. 17	Yellow River Basin
8	Xinluzao No. 9	Inland Northwest	58	Xinluzhong No. 12	Inland Northwest	108	Lumian No. 28	Yellow River Basin
9	Xinluzao No. 10	Inland Northwest	59	Xinluzhong No. 14	Inland Northwest	109	Lu 24	Yellow River Basin
10	Xinluzao No. 11	Inland Northwest	60	Xinluzhong No. 15	Inland Northwest	110	Lu 25	Yellow River Basin
11	Xinluzao No. 12	Inland Northwest	61	Xinluzhong No. 16	Inland Northwest	111	Lu 34	Yellow River Basin
12	Xinluzao No. 13	Inland Northwest	62	Xinluzhong No. 17	Inland Northwest	112	Yumian No. 17	Yangtze River Basin
13	Xinluzao No. 15	Inland Northwest	63	Xinluzhong No. 19	Inland Northwest	113	Baimian No. 1	Yellow River Basin
14	Xinluzao No. 16	Inland Northwest	64	Xinluzhong No. 21	Inland Northwest	114	Zhong 93001	Yellow River Basin
15	Xinluzao No. 17	Inland Northwest	65	Xinluzhong No. 22	Inland Northwest	115	CCRI No. 12	Yellow River Basin
16	Xinluzao No. 18	Inland Northwest	66	Xinluzhong No. 26	Inland Northwest	116	CCRI No. 16	Yellow River Basin
17	Xinluzao No. 19	Inland Northwest	67	Xinluzhong No. 27	Inland Northwest	117	Zhongmian 41	Yellow River Basin
18	Xinluzao No. 20	Inland Northwest	68	Xinluzhong No. 28	Inland Northwest	118	CCRI No. 43	Yellow River Basin
19	Xinluzao No. 21	Inland Northwest	69	Xinluzhong No. 30	Inland Northwest	119	Emian No. 10	Yangtze River Basin
20	Xinluzao No. 23	Inland Northwest	70	Xinluzhong No. 32	Inland Northwest	120	Emian No. 12	Yangtze River Basin
21	Xinluzao No. 24	Inland Northwest	71	Xinluzhong No. 34	Inland Northwest	121	Emian No. 21	Yangtze River Basin
22	Xinluzao No. 25	Inland Northwest	72	Xinluzhong No. 35	Inland Northwest	122	Wanmian 8407	Yangtze River Basin
23	Xinluzao No. 26	Inland Northwest	73	Xinluzhong No. 36	Inland Northwest	123	Sumian No. 8	Yangtze River Basin
24	Xinluzao No. 27	Inland Northwest	74	Xinluzhong No. 38	Inland Northwest	124	Sumian No. 12	Yangtze River Basin
25	Xinluzao No. 29	Inland Northwest	75	Xinluzhong No. 39	Inland Northwest	125	Suyuan 04-129	Yangtze River Basin
26	Xinluzao No. 30	Inland Northwest	76	Xinluzhong No. 40	Inland Northwest	126	Ganmian No. 10	Yangtze River Basin
27	Xinluzao No. 31	Inland Northwest	77	Xinluzhong No. 41	Inland Northwest	127	Ganmian No. 17	Yangtze River Basin
28	Xinluzao No. 32	Inland Northwest	78	Xinluzhong No. 42	Inland Northwest	128	Chuan 7327 20	Yangtze River Basin
29	Xinluzao No. 33	Inland Northwest	79	Xinluzhong No. 44	Inland Northwest	129	Yumian No. 1	Yangtze River Basin
30	Xinluzao No. 34	Inland Northwest	80	Xinluzhong No. 45	Inland Northwest	130	Liaomian No. 9	Special precocious cotton area
31	Xinluzao No. 35	Inland Northwest	81	Xinluzhong No. 46	Inland Northwest	131	Liaomian No. 16	Special precocious cotton area
32	Xinluzao No. 36	Inland Northwest	82	Xinluzhong No. 50	Inland Northwest	132	Dai-80	America
33	Xinluzao No. 37	Inland Northwest	83	Xinluzhong No. 54	Inland Northwest	133	Montenegro cotton No. 1	Special precocious cotton area
34	Xinluzao No. 38	Inland Northwest	84	Xinluzhong No. 56	Inland Northwest	134	Pidcotton	Yellow River Basin
35	Xinluzao No. 39	Inland Northwest	85	Xinluzhong No. 58	Inland Northwest	135	MacNair 210	America
36	Xinluzao No. 40	Inland Northwest	86	Xinluzhong No. 59	Inland Northwest	136	Huazhong 106	Yangtze River Basin
37	Xinluzao No. 41	Inland Northwest	87	Xinluzhong No. 60	Inland Northwest	137	Keyuan No. 1	Yellow River Basin
38	Xinluzao No. 42	Inland Northwest	88	Xinluzhong No. 61	Inland Northwest	138	Bu 3363	America
39	Xinluzao No. 45	Inland Northwest	89	Xinluzhong No. 62	Inland Northwest	139	Bamian No. 1	Yangtze River Basin
40	Xinluzao No. 46	Inland Northwest	90	Xinluzhong No. 63	Inland Northwest	140	Chad No. 3	Africa
41	Xinluzao No. 47	Inland Northwest	91	Xinluzhong No. 64	Inland Northwest	141	Turkmen upland cotton	Central Asia
42	Xinluzao No. 48	Inland Northwest	92	Xinluzhong No. 65	Inland Northwest	142	NO Phenphenol phenol No. 1	Yangtze River Basin
43	Xinluzao No. 49	Inland Northwest	93	Xinluzhong No. 68	Inland Northwest	143	Miscot7803-52	America
44	Xinluzao No. 50	Inland Northwest	94	Xinluzhong No. 69	Inland Northwest	144	Si-6524	Central Asia
45	Xinluzao No. 51	Inland Northwest	95	Xinlu 201	Inland Northwest	145	Sparculent H10	America
46	Xinluzao No. 52	Inland Northwest	96	Xinlu 202	Inland Northwest	146	Yinmian No. 1	Yellow River Basin
47	Xinluzao No. 53	Inland Northwest	97	Nongken No. 5	Inland Northwest	147	America 28114-313	America
48	Xinluzao No. 60	Inland Northwest	98	Shache Soil cotton	Inland Northwest	148	Columbia	South America
49	Xinluzao No. 61	Inland Northwest	99	Kuche T94-4	Inland Northwest	149	Miscot 8711ne	America
50	Xinluzhong No. 1	Inland Northwest	100	Bazhou 6510	Inland Northwest			

**Table 7 plants-11-00097-t007:** Salt tolerance traits and salt tolerance index.

Salt Tolerance Traits	Salt Tolerance Index
7d Germination rate	3d Germination potential index
7d Germination weight	7d Germination rate index
3d Radicle length	7d Radicle weight index
7d Radicle length	3d Radicle length index
Relative germination potential	7d Radicle length index
Relative germination rate	
7d Radicle weight drop rate	
3d Germination potential	
7d Radicle length drop rate	
3d Radicle length drop rate	

## Data Availability

Data sharing is not applicable to this article as no new data were created or analyzed in this study. The file with the vcf with SNPs and the file with phenotyping have been uploaded as attachments.

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
