# Peer review of "Genome-Wide Association Analysis of Salt-Tolerant Traits in Terrestrial Cotton at Seedling Stage"

_plants, 2021, doi:10.3390/plants11010097_

Round 1
Reviewer 1 Report
Soil salinization is one of the big threats to agriculture. Zheng et al. used genome-wide association analysis (GWAS) to associate salt-tolerant traits with different cotton genotypes. Overall, the study is interesting, and the methods used are standardised. The findings may be useful for downstream analysis and breeding. However, I have some concerns regarding the current manuscript.
- L57, Page 2: ‘total of 54,588 … were identified’ -- Sounds the 54,588 SNPs are all related to the 10 salt tolerance traits. Is this correct?
- The authors may discuss or explain a bit that why they selected cotton samples at the seeding stage.
- L69, Page 2: ‘Breeding salt-tolerant crop varieties is the only way to achieve sustainable agricultural development in the future’ -- Is this correct? May reword
- The authors may check and list the version of the bioinformatics tools used.
- The authors may need to expand the legend of all figures as the current ones are less informative.
- The resolution of all figures needs improving
- During PCA analysis, do the authors have any information regarding the cotton samples in the two subgroups, for instance, do the cotton genotypes share similar features in each subgroup? Does the classification have biological meanings?
- During heterozygosity checking, how did the authors resolve possible misalignment caused by homoeologous exchanges (HEs)? As HEs may falsely introduce some heterozygous sites.
- During the kinship checking, are the samples in this study consistent with the one reported in PCA analysis?
- L221, Page 8: “Slightly higher than the previous study” -- The authors may want to further discuss this and give some possible explanation
- Figure 4: The figure is not easy to read. May use ‘popLDdecay’ to check the LD decay and draw the figure
- L247, Page 10 “sites related to 7d_Germination rate 3” -- Please check if this is correct.
- The discussion and conclusion parts need comprehensively improving, as the current version mainly repeats results.
- Please check if the citation format is correct when starting with a person name, for instance, Cai et al. at L54, Page 2.
Author Response
Dear Reviewer:
Thank you for your letter and the reviewers’ comments on our manuscript entitled “Genome-wide association analysis of salt-tolerant traits in terrestrial cotton seedling stage” (ID: plants-1437516). Those comments are very helpful for revising and improving our paper, as well as the important guiding significance to other research. We have studied the comments carefully and made corrections which we hope meet with approval. The main corrections are in the manuscript and the responds to the reviewers’ comments are as follows (the replies are highlighted in blue ).
According to the reviewers’ detailed suggestions, we have made a careful revision on the original manuscript. All revised portions are marked in red in the revised manuscript which we would like to submit for your kind consideration.
Replies to the reviewers’ comments:
Reviewer : Soil salinization is one of the big threats to agriculture. Zheng et al. used genome-wide association analysis (GWAS) to associate salt-tolerant traits with different cotton genotypes. Overall, the study is interesting, and the methods used are standardised. The findings may be useful for downstream analysis and breeding. However, I have some concerns regarding the current manuscript.
- L57, Page 2: ‘total of 54,588 … were identified’ -- Sounds the 54,588 SNPs are all related to the 10 salt tolerance traits. Is this correct?
Response:It was modified in the manuscript.(line 64)
- The authors may discuss or explain a bit that why they selected cotton samples at the seeding stage.
Response:In recent years, with increasing rainwater and serious secondary salinization,in the cotton sowing season, aggravate the secondary salination of cotton seedling period, which will directly affect the survival rate of cotton, it is crucial to study the salt tolerance traits of cotton seedling stage.
3.L69, Page 2: ‘Breeding salt-tolerant crop varieties is the only way to achieve sustainable agricultural development in the future’ -- Is this correct? May reword
Response:It was modified in the manuscript.(line 89-91)Cultivation of salt-tolerant crop varieties is one of the important ways to achieve sustainable agricultural development in the future.
4.The authors may check and list the version of the bioinformatics tools used.
Response:It was modified in the manuscript.
5.The authors may need to expand the legend of all figures as the current ones are less informative.
Response:It was modified in the manuscript.
6.The resolution of all figures needs improving
Response:It was modified in the manuscript.
7.During PCA analysis, do the authors have any information regarding the cotton samples in the two subgroups, for instance, do the cotton genotypes share similar features in each subgroup? Does the classification have biological meanings?
Response:It was modified in the manuscript.(line 122-130)
8.During heterozygosity checking, how did the authors resolve possible misalignment caused by homoeologous exchanges (HEs)? As HEs may falsely introduce some heterozygous sites.
Response:It was modified in the manuscript.(line 137-139)
9.During the kinship checking, are the samples in this study consistent with the one reported in PCA analysis?
Response:It's the same.
10.L221, Page 8: “Slightly higher than the previous study” -- The authors may want to further discuss this and give some possible explanation
Response:It was modified in the manuscript.(line 156-158)
11.Figure 4: The figure is not easy to read. May use ‘popLDdecay’ to check the LD decay and draw the figure
Response:It was modified in the manuscript.
12.L247, Page 10 “sites related to 7d_Germination rate 3” -- Please check if this is correct.
Response:It was modified in the manuscript.(line 181)
- The discussion and conclusion parts need comprehensively improving, as the current version mainly repeats results.
Response:It was modified in the manuscript.
14.Please check if the citation format is correct when starting with a person name, for instance, Cai et al. at L54, Page 2.
Response:It was modified in the manuscript.
Once again, thank you very much for your constructive comments and suggestions which would help us both in English and in depth to improve the quality of the paper.
Kind regards,
Ze-Liang Zhang
E-mail: zzldeyouxiang@126.com
Corresponding author : Xue-Yuan Li
E-mail address: xjmh2338@163.com

Reviewer 2 Report
The GWAS is a powerful approach for identification of genomic regions associated with the economically important traits. In general, the authors identified two candidate genes that are likely involved in the salt tolerance of cotton and provide information on genetic structure of analyzed population. In my opinion, although the main goal of analysis is of high importance, the results are based on low number of individuals and markers and the correction for significance of p-value is not provided. The population used is not described, the information about relationship between their grouping and origin is missing. Introduction and discussion are written sketchily, based on few, not recent publications. I got the impression that publications were selected to slightly obscure of the fact that the number of plants and SNPs analyzed is rather low. Using a small number of plants does not abolish the result, but requires discussing the weaknesses that result from it. Those two sections must be significantly improved. Although I I don't feel qualified to judge about the English language and style I see that the text is written carelessly and some parts are difficult to understand. The language correction is required.
Detailed major and minor comments are listed below.
Detailed major and minor comments are listed below.
Abstract:
Minor comments:
p.1,row.19-20 and p.1, row 24: 137 and 149 accessions are reported. What is the correct number?
I suggest to use codes for analyzed traits or remove the '_' from names.
p.1,row.34: What was detected, alleles or genes?
Introduction:
Major comments:
The introduction is superficial and misses some new results such us:
Sun, Z., Li, H., Zhang, Y., Li, Z., Ke, H., Wu, L., ... & Ma, Z. (2018). Identification of SNPs and candidate genes associated with salt tolerance at the seedling stage in cotton (Gossypium hirsutum L.). Frontiers in plant science, 9, 1011.
Xu, P., Guo, Q., Meng, S., Zhang, X., Xu, Z., Guo, W., & Shen, X. (2021). Genome-wide association analysis reveals genetic variations and candidate genes associated with salt tolerance related traits in Gossypium hirsutum. BMC genomics, 22(1), 1-14.
Abdelraheem, A., Thyssen, G. N., Fang, D. D., Jenkins, J. N., McCarty, J. C., Wedegaertner, T., & Zhang, J. (2021). GWAS reveals consistent QTL for drought and salt tolerance in a MAGIC population of 550 lines derived from intermating of 11 Upland cotton (Gossypium hirsutum) parents. Molecular Genetics and Genomics, 296(1), 119-129.
Many other research were recently published and should be cited.
Minor comments:
p.1,row.44-47: The sentence should be rewritten., especially part "and mining excellent salt-tolerant and salt-tolerant genes".
p.1,row.56-58: "The chip was utilized to perform a GWAS analysis of 288 upland cotton materials, and total of 54,588 SNPs related to 10 salt tolerance traits were identified, of which 8 SNPs were significantly associated with 3 salt tolerance traits (Cai et al, 2017)". The sentence is confusing, all 54,588 SNPs were related to salt tolerance traits? It should by rewritten.
p.1,row.64-65: "Haplotypes and these results indicate the important role of population genetic methods in the selection of genomic regional variation in the process of cotton domestication (Paterson et al, 2012)". It is difficult to understand what the authors meant. Should be rewritten.
Materials and methods:
Major comments:
According to the Plants author guidelines the Materials and methods should be after Results and Discussion.
What was the number of analyzed cultivars? In the Introduction two numbers, 137 and 149, are provided while according to Materials and methods 144 cultivars were sampled but 149 are listed in a Table 1.
Was the file with SNPs filtered? Provide parameters. What was the frequency of missing data?
2.3. Molecular genetic diversity and phylogenetic analyses
- Provide details on phylogenetic analysis (which program in the PHYLIP package was used).
- It is not important which text editor was used to modify the input file but how the file was modified.
- Which software was used to calculate the NJ tree?
- How the reliability of phylogenetic tree was calculated (software, number of replicates used, it is possible to calculate it in PHYLIP using SEQBOOT+CONSENSE), the bootstrap values should by calculated and shown on a tree.
2.4. Population structure and kinship analysis
- Provide number of SNPs used for both analysis, was the file filtered before analysis? Provide parameters for the STRUCTURE (e.g. model, length of burn-in and iterations).
2.5. Linkage disequilibrium analysis
- The only valid information is that it was done using Plink2 (row: 121), please provide information if all SNPs were used for LD, if not how the file was filtered.
2.6. Association analysis of salt tolerance traits
- 4.row.123: "Linkage disequilibrium analysis of natural populations was used to evaluate traits". Mayby to identify SNPs associated with traits?
- Header of Table 2 is missing. This table contains only list of traits, it can be removed as below is 'name' oftrait and its description. Please, describe how each trait/index was measured/calculated. This paragraph is chaotic and difficult to follow.
- Provide parameters for BLUP (default?).
- Population structure and kinship described in the 2.4. There is no need to write it again.
- Description of glm and mlm provided while four models used. Provide information about cmlm, and fastlmm.
- How the salt stress cotton transcriptional group data are related to the GWAS?
Minor:
- 3.row.85-87: well-developed seed or seedlings?
- 3-4,row.90-92. Please clarify what "with a distance between two significant association sites of less than 10 cm means. What is 'cm' ?
- 4.row:107-121: This is not a description of the method used.
Results and analysis
Major comments:
- It should be Results or Results and Discussion (according to the authors guidelines: https://www.mdpi.com/journal/plants/instructions)
- 6.row 190-191:" and there was gene exchange between the two subgroup materials." How the authors reach such conclusion? Explain it to the readers providing exact result supporting this conclusion. Provide the bootstrap.
- 6.row.192-198 - should be moved to Materials and methods
- 6.row198-199 - What was the Q value for each of reported k? It is not clear why grouping for k=2 is 'better' than for K=6. The same is true for NJ tree and PCA. Two groups are not obvious. Describe materials/accessions belonging to each group, provide any characterization showing biological sense in such grouping.
- 8.row.230-231: Was the normal distribution tested using any statistical approach? Replace information that R language was used to test the significance of correlation by the name of a package used. Also, provide the p-value.
- 8,row.233-236. The conclusion that "the correlations between different traits is low, which may be because these 5 traits are con trolled by independent inherited genetic sites in response to salt stress, indicating consistent complexity" is not well supported by the date. Please explain why indexes that are likely (clear description not provided in the materials and Methods) calculated from the same measurable traits should not be correlated. Also, i am not convinced that those 5 traits are controlled by independent genetic sites.
- How the p-value was corrected, why the -log10(p)=3 was used as a cut-off? It seems to be very low.
- 9.row.245: Tables 2-12 are not available.
- Table 4: The Threshold column should be named -log10(p), p-value column is not required. The correction and corrected p-value cutoff should be provided. Otherwise it seems that associations may not be significant.
- Table 5: replace p-value by the -log10(p), provide corrected p-value cut-off.
Minor comments:
Rephrase the Figure 1 caption. For example, (A) is not a construction of tree but NJ tree constructed based, (B) is a scatter plot of principal component analysis (PCA), (C) Estimated population structure, what is E1,E2,..E-6?
Rephrase p.6.row: 221-223.
Qality of Figure 3 is low.
Quality of Figure 5 is low, the subtitles/axis titles in the figure are illegible. Also, describe what is shown in A, B and C. I would suggest to show histogram with a distribution.
Table 3. Include information that Table presents correlation coefficient.
Dissussion:
Results should be discussed with a most recreant publications.
Also, the number of individuals used is very low as well as number of SNPs and the impact of low number of individuals and SNPs on results should be discussed.
What is the relation between genes/ loci identified in this study and previous results of genomic regions of cotton that are associated with the salt tolerance?

Author Response
Dear Reviewer:
Thank you for your letter and the reviewers’ comments on our manuscript entitled “Genome-wide association analysis of salt-tolerant traits in terrestrial cotton seedling stage” (ID: plants-1437516). Those comments are very helpful for revising and improving our paper, as well as the important guiding significance to other research. We have studied the comments carefully and made corrections which we hope meet with approval. The main corrections are in the manuscript and the responds to the reviewers’ comments are as follows (the replies are highlighted in blue ).
According to the reviewers’ detailed suggestions, we have made a careful revision on the original manuscript. All revised portions are marked in red in the revised manuscript which we would like to submit for your kind consideration.
Replies to the reviewers’ comments:
Reviewer : The GWAS is a powerful approach for identification of genomic regions associated with the economically important traits. In general, the authors identified two candidate genes that are likely involved in the salt tolerance of cotton and provide information on genetic structure of analyzed population. In my opinion, although the main goal of analysis is of high importance, the results are based on low number of individuals and markers and the correction for significance of p-value is not provided. The population used is not described, the information about relationship between their grouping and origin is missing. Introduction and discussion are written sketchily, based on few, not recent publications. I got the impression that publications were selected to slightly obscure of the fact that the number of plants and SNPs analyzed is rather low. Using a small number of plants does not abolish the result, but requires discussing the weaknesses that result from it. Those two sections must be significantly improved. Although I I don't feel qualified to judge about the English language and style I see that the text is written carelessly and some parts are difficult to understand. The language correction is required.
Detailed major and minor comments are listed below.
- 1,row.19-20 and p.1, row 24: 137 and 149 accessions are reported. What is the correct number?
Response:It was modified in the manuscript.(line 20-22).137 copies of Chinese cultivated land cotton varieties and 12 land cotton varieties collected from around the world.
2.I suggest to use codes for analyzed traits or remove the '_' from names.
Response:It was modified in the manuscript.The '_' has been removed from the article.
3.p.1,row.34: What was detected, alleles or genes?
Response:Alleles.
- The introduction is superficial and misses some new results such us:
Response:It was modified in the manuscript.(line 46-48,56-61).
5.p.1,row.44-47: The sentence should be rewritten., especially part "and mining excellent salt-tolerant and salt-tolerant genes".
Response:It was modified in the manuscript.(line 46-48).
6.p.1,row.56-58: "The chip was utilized to perform a GWAS analysis of 288 upland cotton materials, and total of 54,588 SNPs related to 10 salt tolerance traits were identified, of which 8 SNPs were significantly associated with 3 salt tolerance traits (Cai et al, 2017)". The sentence is confusing, all 54,588 SNPs were related to salt tolerance traits? It should by rewritten.
Response:It was modified in the manuscript.
7.p.1,row.64-65: "Haplotypes and these results indicate the important role of population genetic methods in the selection of genomic regional variation in the process of cotton domestication (Paterson et al, 2012)". It is difficult to understand what the authors meant. Should be rewritten.
Response:It was modified in the manuscript.Delete it from the article.
8.According to the Plants author guidelines the Materials and methods should be after Results and Discussion.
Response:It was modified in the manuscript.
- What was the number of analyzed cultivars? In the Introduction two numbers, 137 and 149, are provided while according to Materials and methods 144 cultivars were sampled but 149 are listed in a Table 1.
Response:It was modified in the manuscript. 137 Chinese cultivars and 12 collected around the world, totaling 149.
10.Was the file with SNPs filtered? Provide parameters. What was the frequency of missing data?
Response:It was modified in the manuscript.(line 366-368)The SNP data of 149 experimental materials were detected by Illumina Cotton SNP 70K and filtered according to the minor allele frequency (MAF: 0.05) and site integrity (INT: 0.1).
11.Provide details on phylogenetic analysis (which program in the PHYLIP package was used).
Response:It was modified in the manuscript.
12.Which software was used to calculate the NJ tree? How the reliability of phylogenetic tree was calculated (software, number of replicates used, it is possible to calculate it in PHYLIP using SEQBOOT+CONSENSE), the bootstrap values should by calculated and shown on a tree.
Response:It was modified in the manuscript.(line358-363)PHYLIP (http://evolution.genetics.washington.edu/phylip.html) was used to calculate the genetic distance matrix of the sample, Notepad++ software was used to adjust the genetic distance matrix file into a suitable format. After generating the tree file, iTOL (https://itol.embl.de/) was used to draw the NJ tree diagram. Software was used to construct based on the neighbor-joining method and the p-distance model, and bootstrapping was repeated 1,000 times.
- Provide number of SNPs used for both analysis, was the file filtered before analysis? Provide parameters for the STRUCTURE (e.g. model, length of burn-in and iterations).
Response:It was modified in the manuscript.
14.The only valid information is that it was done using Plink2 (row: 121), please provide information if all SNPs were used for LD, if not how the file was filtered.
Response:It was modified in the manuscript.
15.·p.4.row.123: "Linkage disequilibrium analysis of natural populations was used to evaluate traits". Mayby to identify SNPs associated with traits?
Response:It was modified in the manuscript.(line404-407)
- Header of Table 2 is missing. This table contains only list of traits, it can be removed as below is 'name' oftrait and its description. Please, describe how each trait/index was measured/calculated. This paragraph is chaotic and difficult to follow.
Response:It was modified in the manuscript.(line425)
- Provide parameters for BLUP (default?).
Response:Default
- Population structure and kinship described in the 2.4. There is no need to write it again.
Response:It was modified in the manuscript.
- Description of glm and mlm provided while four models used. Provide information about cmlm, and fastlmm.
Response:It was modified in the manuscript.(line404-407,444-450)
- How the salt stress cotton transcriptional group data are related to the GWAS?
Response:It was modified in the manuscript.(line212-218)
- 3.row.85-87: well-developed seed or seedlings?
Response:5-day-old seedlings.
- 3-4,row.90-92. Please clarify what "with a distance between two significant association sites of less than 10 cm means. What is 'cm' ?
Response:It was modified in the manuscript.(line352-354)
- 4.row:107-121: This is not a description of the method used.
Response:It was modified in the manuscript.
- It should be Results or Results and Discussion (according to the authors guidelines: https://www.mdpi.com/journal/plants/instructions)
Response:It was modified in the manuscript.
- 6.row 190-191:" and there was gene exchange between the two subgroup materials." How the authors reach such conclusion? Explain it to the readers providing exact result supporting this conclusion. Provide the bootstrap.
Response:It was modified in the manuscript.The lax answer has been removed from the text.
26.p.6.row.192-198 - should be moved to Materials and methods
Response:It was modified in the manuscript.
27.p.6.row198-199 - What was the Q value for each of reported k? It is not clear why grouping for k=2 is 'better' than for K=6. The same is true for NJ tree and PCA. Two groups are not obvious. Describe materials/accessions belonging to each group, provide any characterization showing biological sense in such grouping.
Response:It was modified in the manuscript.(line118-130)
- 8.row.230-231: Was the normal distribution tested using any statistical approach? Replace information that R language was used to test the significance of correlation by the name of a package used. Also, provide the p-value.
Response:It was modified in the manuscript.Using the D'test package in the R language, normal distribution is considered for p-value> 0.05.
- 8,row.233-236. The conclusion that "the correlations between different traits is low, which may be because these 5 traits are con trolled by independent inherited genetic sites in response to salt stress, indicating consistent complexity" is not well supported by the date. Please explain why indexes that are likely (clear description not provided in the materials and Methods) calculated from the same measurable traits should not be correlated. Also, i am not convinced that those 5 traits are controlled by independent genetic sites.
Response:It was modified in the manuscript.The lax answer has been removed from the text.
- How the p-value was corrected, why the -log10(p)=3 was used as a cut-off? It seems to be very low.
Response:It was modified in the manuscript.(line194-200)According to the Bonferroni correction principle, -log10 (P) > 3.97 (P = 1/n, n is the SNP numbers in this study) should be as threshold, but the Bonferroni correction is too stringent, we can't almost identify the significant SNPs for two traits with this threshold. To obtain more associated SNPs, the significantly associated SNP markers with salt-tolerant-relatedvtraits were identified according too -log10 P > 3.0 (Hatzig et al., 2015; Zhang et al., 2015a).
- 9.row.245: Tables 2-12 are not available.
Response:It was modified in the manuscript.
- Table 4: The Threshold column should be named -log10(p), p-value column is not required. The correction and corrected p-value cutoff should be provided. Otherwise it seems that associations may not be significant.
Response:It was modified in the manuscript.
- Table 5: replace p-value by the -log10(p), provide corrected p-value cut-off.
Response:It was modified in the manuscript.
34.Rephrase the Figure 1 caption. For example, (A) is not a construction of tree but NJ tree constructed based, (B) is a scatter plot of principal component analysis (PCA), (C) Estimated population structure, what is E1,E2,..E-6?
Response:It was modified in the manuscript. Image corrected.
35.Rephrase p.6.row: 221-223. Qality of Figure 3 is low.
Response:It was modified in the manuscript. Image corrected.
- Quality of Figure 5 is low, the subtitles/axis titles in the figure are illegible. Also, describe what is shown in A, B and C. I would suggest to show histogram with a distribution.
Response:It was modified in the manuscript. Image corrected.
- Table 3. Include information that Table presents correlation coefficient.
Response:It was modified in the manuscript. Image corrected.
- Results should be discussed with a most recreant publications.
Response:It was modified in the manuscript.
- Also, the number of individuals used is very low as well as number of SNPs and the impact of low number of individuals and SNPs on results should be discussed.
Response:For GWAS analysis, to increase the sample size is the most direct and effective way to improve test efficiency, and for low-frequency and rare mutation sites, conventional sample sizes could not be effectively detected because samples carrying the corresponding Allel are too few and difficult to achieve statistical significance. Most of the number of samples selected in many other literature was 150 samples.Genetic parameters with different sample sizes had different effects on cotton. Genetic variation, population of cotton groups, and the percentage of polymorphic numbers within populations influenced genetic parameters. Influence from genetic variation and genetic parameters between groups was strong, but total numbers and polymorphic numbers were not affected. The number of samples revealed different ethnogenic relatedness, too small sample numbers, and clustering results to reflect the relatedness of cotton populations.
- What is the relation between genes/ loci identified in this study and previous results of genomic regions of cotton that are associated with the salt tolerance?
Response:It was modified in the manuscript.(line212-218,250-257,279-295)
Once again, thank you very much for your constructive comments and suggestions which would help us both in English and in depth to improve the quality of the paper.
Kind regards,
Ze-Liang Zhang
E-mail: zzldeyouxiang@126.com
Corresponding author : Xue-Yuan Li
E-mail address: xjmh2338@163.com

Reviewer 3 Report
The manuscript presents data on the identification of SNP markers associated with such important economically valuable traits as resistance to soil salinity. For this, the authors have collected an extensive collection of samples of cotton Gossypium hirsutum, represented by 149 samples, most of which grow in China. To identify polymorphism by single nucleotide substitutions, the authors used the technology of DNA chips, which made it possible to identify in the analyzed material SNP loci associated with the resistance of cotton plants to an increased salt content in the early stages of development. The authors correctly selected the early stage of cotton development for analysis, since the resistance to salinity can vary depending on the vegetation phase of the plant and the stage of seed germination is most susceptible to adverse effects during soil salinity. The importance of genome-wide associative analysis for identifying and mapping SNPs associated with economically valuable traits in cotton (yield, fiber quality, etc.), as well as resistance to biotic and abiotic stresses, including salt tolerance, is emphasized in the review by MAJEED Sajid and co-authors, Role of SNPs in determining QTLs for major traits in cotton (Journal of Cotton Research, 2019). Among 18 432 SNP markers identified by the authors, 27 significant SNP sites were associated with the salt tolerance trait, and 15 - with the salt tolerance index. The relationship between significant sites and a decrease in the length of the shoot is shown. The results of an increase in the expression of two identified genes after 12 hours of salt stress are presented. The data obtained by the authors of the manuscript are of interest for further work in the direction of searching for molecular markers associated with salt tolerance in cotton. This manuscript can be recommended for publication in the journal. The reviewer did not find work in the cited literature (Sun et al, 2017) - line 297.
Author Response
Dear Reviewer:
Thank you for your letter and the reviewers’ comments on our manuscript entitled “Genome-wide association analysis of salt-tolerant traits in terrestrial cotton seedling stage” (ID: plants-1437516). Those comments are very helpful for revising and improving our paper, as well as the important guiding significance to other research. We have studied the comments carefully and made corrections which we hope meet with approval. The main corrections are in the manuscript and the responds to the reviewers’ comments are as follows (the replies are highlighted in blue ).
According to the reviewers’ detailed suggestions, we have made a careful revision on the original manuscript. All revised portions are marked in red in the revised manuscript which we would like to submit for your kind consideration.
Replies to the reviewers’ comments:
Reviewer : The manuscript presents data on the identification of SNP markers associated with such important economically valuable traits as resistance to soil salinity. For this, the authors have collected an extensive collection of samples of cotton Gossypium hirsutum, represented by 149 samples, most of which grow in China. To identify polymorphism by single nucleotide substitutions, the authors used the technology of DNA chips, which made it possible to identify in the analyzed material SNP loci associated with the resistance of cotton plants to an increased salt content in the early stages of development. The authors correctly selected the early stage of cotton development for analysis, since the resistance to salinity can vary depending on the vegetation phase of the plant and the stage of seed germination is most susceptible to adverse effects during soil salinity. The importance of genome-wide associative analysis for identifying and mapping SNPs associated with economically valuable traits in cotton (yield, fiber quality, etc.), as well as resistance to biotic and abiotic stresses, including salt tolerance, is emphasized in the review by MAJEED Sajid and co-authors, Role of SNPs in determining QTLs for major traits in cotton (Journal of Cotton Research, 2019). Among 18 432 SNP markers identified by the authors, 27 significant SNP sites were associated with the salt tolerance trait, and 15 - with the salt tolerance index. The relationship between significant sites and a decrease in the length of the shoot is shown. The results of an increase in the expression of two identified genes after 12 hours of salt stress are presented. The data obtained by the authors of the manuscript are of interest for further work in the direction of searching for molecular markers associated with salt tolerance in cotton. This manuscript can be recommended for publication in the journal. The reviewer did not find work in the cited literature (Sun et al, 2017) - line 297.
Response:It was modified in the manuscript.(line 549)Sun Z, Wang X, Liu Z, et al, 2017, Genome—wide association study discovered genetic variation and candidate genes of fibre quality traits in Gossypium hirsutum L. Plant Biotechnol , Aug, 15(8)982-996.
Once again, thank you very much for your constructive comments and suggestions which would help us both in English and in depth to improve the quality of the paper.
Kind regards,
Ze-Liang Zhang
E-mail: zzldeyouxiang@126.com
Corresponding author : Xue-Yuan Li
E-mail address: xjmh2338@163.com
Round 2
Reviewer 1 Report
I'm satisfied with the revision.
By the way, for citations starting with a person name, the authors may need to remove the brackets when proofreading.
Author Response
Dear Reviewer:
Thank you for your letter and the reviewers’ comments on our manuscript entitled “Genome-wide association analysis of salt-tolerant traits in terrestrial cotton seedling stage” (ID: plants-1437516). Those comments are very helpful for revising and improving our paper, as well as the important guiding significance to other research. We have studied the comments carefully and made corrections which we hope meet with approval. The main corrections are in the manuscript and the responds to the reviewers’ comments are as follows (the replies are highlighted in blue ).
According to the reviewers’ detailed suggestions, we have made a careful revision on the original manuscript. All revised portions are marked in red in the revised manuscript which we would like to submit for your kind consideration.
Replies to the reviewers’ comments:
Reviewer : By the way, for citations starting with a person name, the authors may need to remove the brackets when proofreading.
Response:Thanks for your advice. It was modified in the manuscript.
Once again, thank you very much for your constructive comments and suggestions which would help us both in English and in depth to improve the quality of the paper.
Kind regards,
Ze-Liang Zhang
E-mail: zzldeyouxiang@126.com
Corresponding author : Xue-Yuan Li
E-mail address: xjmh2338@163.com

Reviewer 2 Report
Authors tried to address all my comments and suggestions but some sections sill should be improved. The manuscript requires extensive language editing. Lots of sentences are difficult to understand, especially those that were re-written or added to the revised version, for example:
“we analysed the population structure and genetic diversity has a 19 total of 149 materials”,
‘To investigate the expression pattern of these genes during the seedling stage under salt stress tolerance, to further screen the possible candidate genes involved in the salt response, these genes were analyzed using the expression level of the seedlings at 1, 3, 6, and 12 h under 400 mM salt concentration reported public transcriptome data sets were retrieved from ccNET (https://structralbiology.cau.edu.cn.gossypium) (Zhang et al. 211 2015b). ‘ and many other.
Sometimes it seems that part of the sentence is missing: ‘Figure 1. Analysis of group structure. A: NJ tree constructed based. ‘ Based on what?
Some additional citations are added to the introduction, however there is only information about number of markers and plants used for analysis. Important information about significant SNPs and candidate genes (with short description of their function) is missing. Also, if the citation refers to the author only date should be in brackets.
I still think that bootstrap values should be added to the Figure 1, otherwise the figure is not informative.
Table 2: in the column ‘-log10(p) the p value is presented while in the column “Threshold contains –log10(p).
Table 3: in the column ‘-log10(p) the p value is presented.
In the ‘materials and methods’ five models are reported (mlm, glm, cmlm, emmax and fastlmm). Please, provide information about performance of those models, differences in results and information which one was used for the final analysis (to obtain results described in the ‘Results’ section). Were those models used for the association analysis, that is described in a section below?
In the section describing candidate gene screening information about identification of candidate genes is missing. How those genes were selected? How many regions were identified? How many genes were present in identified regions (based on materials and methods: a 500 kb upstream and downstream of the physical location of each SNP site was used as the candidate gene physical location).
The only information is that: ‘Five SNP sites on chromosome A01 and two SNP sites on chromosome D01 associated with a decrease in Radicle length at 7 days were detected. As a result, The alleles Gh_ A01G0034 and Gh_D01G0028 related to salt tolerance were detected and are located in A01 and D01 by homology.’ The sentence is poorly written, it is not clear what is the relation between SNPs and the candidate genes.
Data availability statements: Data sharing is not applicable to this article as no new data were 457 created or analyzed in this study. – If I understand correctly the ‘materials and methods’ the genotyping and phenotyping used for the GWAS was done for this study. If so, the file with the vcf with SNPs and file with phenotyping should be added as a supplementary files. If those are not new data their source should be provided.
Author Response
Dear Reviewer:
Thank you for your letter and the reviewers’ comments on our manuscript entitled “Genome-wide association analysis of salt-tolerant traits in terrestrial cotton seedling stage” (ID: plants-1437516). Those comments are very helpful for revising and improving our paper, as well as the important guiding significance to other research. We have studied the comments carefully and made corrections which we hope meet with approval. The main corrections are in the manuscript and the responds to the reviewers’ comments are as follows (the replies are highlighted in blue ).
According to the reviewers’ detailed suggestions, we have made a careful revision on the original manuscript. All revised portions are marked in red in the revised manuscript which we would like to submit for your kind consideration.
Replies to the reviewers’ comments:
Reviewer : Authors tried to address all my comments and suggestions but some sections sill should be improved. The manuscript requires extensive language editing. Lots of sentences are difficult to understand, especially those that were re-written or added to the revised version, for example: “we analysed the population structure and genetic diversity has a 19 total of 149 materials”,
Response:Thanks for your advice.The original text: we analysed the population structure and genetic diversity has a total of 149 materials of which 137 elite Gossypium hirsutum cultivar accessions collected from China and 12 elite Gossypium hirsutum cultivar accessions collected from around the world,the extra 19 may be the line number.
‘To investigate the expression pattern of these genes during the seedling stage under salt stress tolerance, to further screen the possible candidate genes involved in the salt response, these genes were analyzed using the expression level of the seedlings at 1, 3, 6, and 12 h under 400 mM salt concentration reported public transcriptome data sets were retrieved from ccNET (https://structralbiology.cau.edu.cn.gossypium) (Zhang et al. 211 2015b). ‘ and many other.
Response:Thanks for your advice. We have made the necessary modifications according to your comments and suggestions. We hope that the revised manuscript will be approved for publication.
Sometimes it seems that part of the sentence is missing: ‘Figure 1. Analysis of group structure. A: NJ tree constructed based. ‘ Based on what?
Response: Thanks for your advice. It was modified in the manuscript.
Some additional citations are added to the introduction, however there is only information about number of markers and plants used for analysis. Important information about significant SNPs and candidate genes (with short description of their function) is missing. Also, if the citation refers to the author only date should be in brackets.
Response: Thanks for your advice. It was modified in the manuscript.A brief description of important gene functions is mainly given in Section 3.2.
I still think that bootstrap values should be added to the Figure 1, otherwise the figure is not informative.
Response: Thanks for your advice. We calculated bootstrap values to assess the credibility of the tree with all branches above 50%, but due to more material and too large data to be displayed on the graph due to picture resolution problems, with the picture here as the basis for material grouping only.
Table 2: in the column ‘-log10(p) the p value is presented while in the column “Threshold contains –log10(p). Table 3: in the column ‘-log10(p) the p value is presented.
Response: Thanks for your advice. It was modified in the manuscript.
In the ‘materials and methods’ five models are reported (mlm, glm, cmlm, emmax and fastlmm). Please, provide information about performance of those models, differences in results and information which one was used for the final analysis (to obtain results described in the ‘Results’ section). Were those models used for the association analysis, that is described in a section below?
Response: Thanks for your advice. It was modified in the manuscript. Due to the small number of environments and the existence of certain false positives, the CMLM model can reduce the false positives as much as possible, so the method of CMLM is adopted.
In the section describing candidate gene screening information about identification of candidate genes is missing. How those genes were selected? How many regions were identified? How many genes were present in identified regions (based on materials and methods: a 500 kb upstream and downstream of the physical location of each SNP site was used as the candidate gene physical location).
Response: Thanks for your advice. It was modified in the manuscript.
The only information is that: ‘Five SNP sites on chromosome A01 and two SNP sites on chromosome D01 associated with a decrease in Radicle length at 7 days were detected. As a result, The alleles Gh_ A01G0034 and Gh_D01G0028 related to salt tolerance were detected and are located in A01 and D01 by homology.’ The sentence is poorly written, it is not clear what is the relation between SNPs and the candidate genes.
Response: Thanks for your advice. It was modified in the manuscript.
Data availability statements: Data sharing is not applicable to this article as no new data were 457 created or analyzed in this study. – If I understand correctly the ‘materials and methods’ the genotyping and phenotyping used for the GWAS was done for this study. If so, the file with the vcf with SNPs and file with phenotyping should be added as a supplementary files. If those are not new data their source should be provided.
Response: Thanks for your advice. The file with the vcf with SNPs and file with phenotyping have been uploaded for attachment.
Once again, thank you very much for your constructive comments and suggestions which would help us both in English and in depth to improve the quality of the paper.
Kind regards,
Ze-Liang Zhang
E-mail: zzldeyouxiang@126.com
Corresponding author : Xue-Yuan Li
E-mail address: xjmh2338@163.com

Round 3
Reviewer 2 Report
The authors responded to all my comments and dispel my doubts. In my opinion, the manuscript is acceptable for publication in its present form.